# Predictive Modeling of Alzheimer’s and Parkinson’s Disease Using Metabolomic and Lipidomic Profiles from Cerebrospinal Fluid

**DOI:** 10.3390/metabo12040277

**Published:** 2022-03-22

**Authors:** Nathan Hwangbo, Xinyu Zhang, Daniel Raftery, Haiwei Gu, Shu-Ching Hu, Thomas J. Montine, Joseph F. Quinn, Kathryn A. Chung, Amie L. Hiller, Dongfang Wang, Qiang Fei, Lisa Bettcher, Cyrus P. Zabetian, Elaine R. Peskind, Ge Li, Daniel E. L. Promislow, Marie Y. Davis, Alexander Franks

**Affiliations:** 1Department of Statistics and Applied Probability, University of California, Santa Barbara, CA 93106, USA; afranks@pstat.ucsb.edu; 2Northwest Metabolomics Research Center, Department of Anesthesiology and Pain Medicine, University of Washington School of Medicine, Seattle, WA 98195, USA; xinyuzche@gmail.com (X.Z.); draftery@uw.edu (D.R.); haiweigu@asu.edu (H.G.); dfwang@hospital.cqmu.edu.cn (D.W.); feiqiang@gmail.com (Q.F.); lisafanbettcher@gmail.com (L.B.); 3Veterans Affairs Puget Sound Health Care System, Seattle, WA 98108, USA; shuching@uw.edu (S.-C.H.); zabetian@uw.edu (C.P.Z.); peskind@uw.edu (E.R.P.); gli@uw.edu (G.L.); myd@uw.edu (M.Y.D.); 4Department of Neurology, University of Washington School of Medicine, Seattle, WA 98195, USA; 5Department of Pathology, Stanford University School of Medicine, Palo Alto, CA 94304, USA; tmontine@stanford.edu; 6Portland Veterans Affairs Medical Center, Portland, OR 97239, USA; quinnj@ohsu.edu (J.F.Q.); chungka@ohsu.edu (K.A.C.); peterami@ohsu.edu (A.L.H.); 7Department of Neurology, Oregon Health and Science University, Portland, OR 97239, USA; 8Department of Psychiatry and Behavioral Sciences, University of Washington School of Medicine, Seattle, WA 98102, USA; 9Department of Biology, University of Washington, Seattle, WA 98105, USA; promislo@uw.edu; 10Department of Laboratory Medicine & Pathology, University of Washington School of Medicine, Seattle, WA 98195, USA

**Keywords:** predictive modeling, biomarker, cerebrospinal fluid, cross-sectional study, neurodegenerative disease

## Abstract

In recent years, metabolomics has been used as a powerful tool to better understand the physiology of neurodegenerative diseases and identify potential biomarkers for progression. We used targeted and untargeted aqueous, and lipidomic profiles of the metabolome from human cerebrospinal fluid to build multivariate predictive models distinguishing patients with Alzheimer’s disease (AD), Parkinson’s disease (PD), and healthy age-matched controls. We emphasize several statistical challenges associated with metabolomic studies where the number of measured metabolites far exceeds sample size. We found strong separation in the metabolome between PD and controls, as well as between PD and AD, with weaker separation between AD and controls. Consistent with existing literature, we found alanine, kynurenine, tryptophan, and serine to be associated with PD classification against controls, while alanine, creatine, and long chain ceramides were associated with AD classification against controls. We conducted a univariate pathway analysis of untargeted and targeted metabolite profiles and find that vitamin E and urea cycle metabolism pathways are associated with PD, while the aspartate/asparagine and c21-steroid hormone biosynthesis pathways are associated with AD. We also found that the amount of metabolite missingness varied by phenotype, highlighting the importance of examining missing data in future metabolomic studies.

## 1. Introduction

Alzheimer’s (AD) and Parkinson’s (PD) disease are the two most common neurodegenerative disorders, and the fifth and tenth leading causes of death in the United States among individuals aged 65 or older [1]. Projections using the 2010 US census estimate that there will be approximately 1.2 million cases of PD (among individuals aged 45 or older) and 8.4 million cases of AD by 2030 [2,3]. Currently, there are no therapies that can slow or halt progression of these debilitating diseases. Biomarkers for these diseases could improve our ability to diagnose these conditions at early stages, and track progression of neurodegeneration and the effect of therapies on slowing or halting disease progression.

Recent research has focused on the discovery of biomarkers to identify the presence of these diseases [4,5]. Cerebrospinal fluid (CSF) surrounds the brain and is peripherally accessible, making it a promising medium to use for identifying biomarkers of neurodegeneration in AD and PD [6,7]. The metabolome, consisting of the small molecules (<2000 Da) circulating in an organism, has been shown to vary across many phenotypic traits, including (but not limited to) human longevity [8], cancer [9,10], and type 2 diabetes [11]. Alterations in the metabolome have also been identified in AD versus control subjects [12,13], and PD versus control subjects [14,15,16].

In the present study, we examined whether the CSF metabolome can discriminate between healthy controls, AD, and PD subjects by fitting predictive models for each disease. We compared untargeted aqueous, targeted aqueous, and lipidomic approaches to profiling the metabolome in their ability to discriminate between AD, PD, and control CSF. Using a logistic elastic net regression to classify phenotypes and select relevant features [17], we were able to best discriminate PD versus control in the untargeted metabolite profiles (0.998 AUC). Our models were also able to discriminate AD versus control in untargeted and lipid profiles well (0.81 and 0.75 AUC, respectively). Our model’s ability to differentiate between AD and PD in the untargeted metabolomic profile was also excellent (0.976 AUC). We performed univariate analyses to identify potential metabolomic pathways associated with each disease, revealing metabolites and lipids that have been previously identified in association with AD and PD versus controls, as well as some new putative biomarkers. We address statistical methods for handling potential data quality issues related to metabolomics, including corrections for assay drift and an examination of statistically informative missingness. Our work should be validated in follow-up studies to determine if our findings are robust in larger cohorts, and in longitudinal studies to identify if any metabolites are biomarkers for disease progression. To our knowledge, this is the first paper to compare multivariate classification performance of AD and PD between metabolomic profiles.

## 2. Results

CSF samples from AD, PD, and healthy control subjects obtained through two different longitudinal studies were analyzed in this study: 85 healthy subjects, 57 subjects with AD, and 56 subjects with PD. Age at time of CSF collection ranged from 20 to 88, with a median age of 65 (Table 1).

The average age of control subjects was 54 at the time of LP, ranging from 20 to 86 years old, with 48% female and 93% white. The average age of AD subjects at the time of LP was 71, ranging from 52 to 87 years old, with 51% female and 95% white. The average duration of disease of AD subjects was 4.3±2.8 years. Consistent with the diagnosis of AD, the average MMSE score for AD subjects was 21.4±5.6. compared to 29.4±0.9 for controls. Additional cognitive measures for AD and control subjects are included in Table 1.

The average age of the PD subjects was 65, with 9 years average duration of disease at the time of LP. The average levodopa equivalent daily dose (LEDD) was 714 mg/day, where only three PD subjects were not taking dopaminergic medication. The average MDS-UPDRS part III score for motor symptoms was 25, with average Hoehn and Yahr stage of 2, consistent with mid-stage PD patients on a moderate amount of dopaminergic medication. The average MOCA score for the PD cohort was 25.2 ± 3.8. The majority of the cohort had the consensus diagnosis of mild cognitive impairment (64%), while 29% had no cognitive impairment and 7% had dementia at the time of LP. Six PD subjects were carriers of pathogenic GBA gene mutations and two subjects were carriers of the GBA E326K polymorphism. None of the most common pathogenic mutations of leucine rich repeat kinase 2 (LRRK2) G2019S or R1441C/G/H/S mutation carriers were identified in this cohort.

Metabolomic profiling of CSF was performed using untargeted and targeted metabolite methods, as well as lipid profiling. The untargeted, targeted, and lipid profiling yielded 6735, 108, and 1070 features, respectively. These samples came with widely varying degrees of missingness, with 16%, 3%, and 81% of the data missing from each of the profiles, respectively. A summary of subject age and missing data are displayed in Appendix B Figure A1, a flowchart outlining the analysis performed in this study is available in Appendix C Figure A2, and code used to run the analysis can be found in the Appendix A.

### 2.1. PCA

We first used the unsupervised method of principal component analysis (PCA) (see, e.g., [19]) to identify the combinations of metabolites responsible for the majority of the variation present in the untargeted metabolomic data. Twenty percent of the variation in the data was explained by the first two principal components (Figure 1). This variation did not associate strongly with disease (AD, PD, controls), as the projection of the subjects’ untargeted profile onto the first two principal components does not show significant separation between subject groups. In a previous analysis on the dataset of only control subjects, we found the first principal component to be associated with age [20].

### 2.2. Prediction Results

We next used supervised methods to seek explicit combinations of metabolites and lipids that can differentiate AD, PD, and healthy controls. For both its predictive power and interpretable results, we fit binary logistic elastic net regression models on each of these three pairwise phenotype combinations (AD, PD, control), and on three profiles (untargeted and targeted aqueous metabolites and untargeted lipids), using leave one out prediction to classify the disease status for each subject. The overall predictive performance of models fit on the untargeted profiles is displayed via ROC curves in Figure 2. Alternative characterizations of model performance are presented in Appendix F, including precision–recall curves and examples of threshold-dependent performance metrics. As stated in Section 4.2, age and sex are detrended from each metabolite prior to modeling fitting to better distinguish the signal from the metabolome from the known association of these covariates between AD, PD, and controls. The predictive performance of models fit including age and sex can be found in Appendix G.

#### 2.2.1. Classifying AD against Controls

The predictive accuracy of models classifying AD against controls varied across the profiles, reporting Area Under the ROC Curves (AUC) between 0.5 and 0.7. The untargeted profile yielded the highest predictive accuracy. Recall that age and sex are detrended from each metabolite, so that these models mostly disregard the discriminatory power of age and sex. These models contain less discriminatory power than age and sex alone, as the leave one out performance of classical logistic regression models using only age and sex yields an AUC of 0.734. In addition to the leave one out modeling done to generate these ROC curves, a logistic elastic net model was fit on the full lipid and targeted profiles to get a sense of which metabolites separate AD from controls in our data. Exponentiated coefficients larger than 1.1 or smaller than 0.9 for these models are displayed in Table 2, and represent the odds ratio (OR) of classifying the subject as having AD resulting from a standard deviation increase in the metabolite/lipid.

#### 2.2.2. Classifying PD against Controls

The models correctly classify PD patients with near perfect accuracy in the untargeted metabolomic profile, and classify better than chance in the targeted metabolomic and lipidomic profiles. For reference, the leave one out predictions of logistic regression models fit using only age and sex yield AUC of 0.671. Table 3 displays the odds ratios associated with an elastic net model fit on the full targeted metabolite and lipid profiles, akin to Table 2. To test whether this near perfect prediction in the untargeted profile can be attributed to a small subset of metabolites, we proceeded to run a second elastic net regression, following the same procedure described in Section 4.3, but this time removing all metabolites selected by the first elastic net regression model from consideration (fit using all controls and PD subjects). This second elastic net model was still predictive of PD, reporting an average AUC of 0.83 over the five imputations and the leave one out analysis, indicating that the discriminatory power of our model is not limited to a small set of metabolites.

Mutations in GBA are the strongest genetic risk factor for PD, increasing the risk of PD in heterozygous carriers by approximately 5-fold compared to non-carriers [21], and are present in  5% of all PD patients. GBA E326K is also associated with PD risk, albeit with a smaller effect size [22], and both GBA mutations and E326K are associated with more rapid progression of cognitive and motor symptoms in PD [23,24,25]. Our cohort of PD patients included six GBA mutation carriers and two GBA E326K carriers. Due to the small size of both genotype groups, we pooled the six mutation carriers and two E326K carriers to improve power for our analysis. Models classifying the eight GBA variant carriers against non-carriers within the PD cohort do not perform better than chance. We also examined whether PD-related medications could be significantly influencing metabolic profiles, as 53 of the 56 PD subjects were taking dopaminergic medication at the time of LP. However, models predicting LEDD from metabolites had very little predictive power, not performing much better than a naive intercept-only model. Additionally, we performed a reanalysis of the untargeted profile after removing all metabolites within 1 *m*/*z* of levodopa, entacapone, and their drug metabolites (dopamine, dihydroxyphenylacetic acid, homovanillic acid, and 3-*O*-methyldopa, *S*-adenosyl-l-methionine, norepinephrine, epinephrine). In this reanalysis, we found no reduction in the discriminative performance of our models.

#### 2.2.3. Classifying AD from PD

In addition to classifying disease subjects versus controls, models were also fit to classify AD versus PD using the metabolome. Similar to PD versus control classifications, the untargeted profile was able to distinguish between the two classes with near perfect accuracy, while the lipid models do not perform significantly better than chance. For reference, the leave one out predictions of logistic regression models using only age and sex yields an AUC of 0.716. The most significant metabolites distinguishing between AD and PD, as determined by analyzing untargeted metabolite profiles, are summarized in Table 4. The corresponding tables for lipids are reported in Appendix J Table A3.

### 2.3. Missing Data

We noted that there was a significant percentage of missing features for targeted and untargeted aqueous metabolite and lipid profiles per group. To determine whether there could be any correlation between the pattern of missing features and AD, PD, or controls, we fit a series of models following the procedure described in Section 4.3 with the modifications that all metabolite abundances were replaced with 1’s if they were present, and 0 if they were missing. Age and sex were still included in this model to control for their possible associations with missingness [20]. These matrices of missingness indicators (along with sex at birth and age) were still moderately to extremely successful at classifying AD and PD against controls, reporting leave one out AUCs as high as 0.97 (classifying PD against controls in the untargeted profile). The high predictive accuracy at classifying PD against controls in the untargeted missingness indicator profile is moderately robust, as removing all significant metabolites and rerunning the analysis yields an AUC of 0.72. However, the near perfect classification result reported in Section 2.2.2 is not solely a result of differential missingness, as similar performance is achieved after removing all metabolites that were missing in more than 1% of samples. Targeted and lipidomic tables for the full models (following the same format as Table 2 and Table 3) are shown in Table 5.

We also found that, when the proportion of missing data between phenotype subjects is large, the controls tended to have the most missingness, and PD subjects have the least missingness (after controlling for age). A series of univariate binary logistic regressions were fit on the untargeted profile to classify each metabolite’s pattern of abundance missingness based on phenotype (AD, PD, or age-matched controls). All metabolites with at least one coefficient FDR <0.05 are displayed in Appendix E Figure A4.

### 2.4. Pathway and Set Enrichment Analysis

We used pathway analysis to map significant individual metabolites identified by our analysis to known metabolite networks. To annotate the metabolites obtained in an untargeted method, we used mummichog, which combines the tasks of pathway analysis and metabolite identification using mass:charge, retention time, MS ion mode, along with univariate classification results. As shown in Figure 3, mummichog identified vitamin B3 (nicotinate and nicotinamide) metabolism and alanine and aspartate metabolite pathways with the negative ion mode metabolites classifying PD against controls. No associations were found classifying AD against controls with negative ion mode metabolites at a false discovery rate <0.05. For the positive ion mode detected metabolites, urea cycle/amino group metabolism, aspartate and asparagine metabolism, C21-steroid hormone biosynthesis and metabolism, biopterin metabolism, glutathione metabolism, vitamin B12 (cyanocobalamin) metabolism, vitamin E metabolism, bile acid biosynthesis, drug metabolism–cytochrome P450, and arginine and proline metabolisms were found to be associated with PD at the false discovery rate <0.05. Classifying AD against controls, mummichog identified the C21-steroid hormone biosynthesis and metabolism, sialic acid metabolism, androgen and estrogen biosynthesis metabolism, starch and sucrose metabolism, and hexose phosphorylation pathways at a false discovery rate <0.05.

Using metabolite abundance missingness indicators to classify PD against controls in univariate analysis, we found vitamin E metabolism to be the only pathway associated with the significant metabolites at a false discovery rate <0.05 in the positive ion mode detected metabolites. No associations were found using the negative mode metabolite missingness to classify PD against controls. No associations were found using positive or negative mode metabolite missingness to classify AD against controls. Using significant metabolites from untargeted univariate analysis classifying AD against PD, mummichog identified the ascorbate (vitamin C) and aldarate metabolism pathways in the negative ion mode, and drug metabolism-cytochrome P450 and vitamin E metabolism pathways in the positive ion mode.

For the targeted profile, where metabolite identities have already been established, we used Metabolite Set Enrichment Analysis (MSEA), which performed overrepresentation analysis to compare significant metabolites from univariate classification against known metabolites sets compiled from literature. MSEA found no significant relationships using *p*-values computed from the hypergeometric distribution. Using metabolite sets from the Small Molecule Pathways Database (SMPDB) [26], we find that the targeted metabolites identified in our univariate classification of PD against controls match two out of three metabolites in the carnitine synthesis set, two out of four of the metabolites in the betaine metabolism set, and three out of eight of the metabolites in the methionine metabolism set. A table of the sets with the largest overlap can be found in Appendix H.

## 3. Discussion

In this study, we analyzed targeted and untargeted metabolomic and lipidomic CSF profiles to develop possible predictive models for AD and PD. Using elastic net regression models, we were able to classify PD against both controls and AD with a high degree of accuracy using untargeted metabolomic profiles, as well as moderate accuracy for all other models classifying phenotype using the targeted and untargeted metabolome. We found that some metabolites driving these predictive multivariate models, such as alanine and ornithine, have already been reported in the existing literature, providing further support for our results. We also performed univariate analysis coupled with pathway and set enrichment analysis and identified putative metabolomic pathways that have been previously associated with AD or PD. Associations between the presence of missing values and phenotype were also reported, suggesting the need for careful treatment of missing data in future metabolomic studies.

While several similar studies analyzing metabolomic and/or lipidomic profiles have been published, our study has several unique strengths. First, we analyzed both targeted and untargeted metabolomic profiles and were able to distinguish a specific metabolomic signature associated with neurodegeneration in both an unbiased interrogation of the metabolome as well as in metabolites of interest. In addition, most prior studies have analyzed metabolomic or lipidomic alterations in a single disease versus controls. We analyzed metabolomic and lipidomic profiles from both AD and PD in relation to each other as well as compared to controls to identify metabolomic and lipidomic changes that might be specific to each disease versus common to neurodegeneration. We were able to distinguish between the metabolome of AD and PD CSF, supporting growing evidence that neurodegenerative diseases have unique metabolomic “signatures”. Finally, we analyzed CSF from living patients rather than from post mortem samples, which may be advantageous in revealing significant alterations in the metabolome. This could be relevant to developing biomarkers of disease progression, rather than alterations that are only detectable in end-stage disease.

The leave-one-out predictive accuracy of our models ranges greatly depending on the classification task and profile used. Models fit using the untargeted profile tended to be the most accurate, classifying PD against controls and PD against AD nearly perfectly, and classifying AD against controls better than chance. We also found that the models achieving near perfect accuracy are not limited to a few metabolites, and that the pattern of missing values alone is enough to distinguish these phenotypes. This suggests that features with missingness may encode important information about phenotype and should be treated with careful consideration in future metabolomics studies. There were a small number of targeted metabolites and lipids that appeared in all leave one out models, which are reported in Appendix K.

Interestingly, the lipidomic profiles of AD and PD subjects were nearly indistinguishable, despite our models being able to classify the lipidomic profiles of PD against controls better than chance. This suggests that there may be common alterations in lipid metabolism in both neurodegenerative diseases. Increasing evidence indicates that alterations in lipid metabolism, particularly ceramide and sphingolipid metabolism, are present in multiple neurodegenerative diseases including AD and PD. This study is one of only a few studies able to compare AD and PD CSF lipidomic profiles obtained at the same time by the same laboratory, reducing alterations due to batch effect. Alterations in lipid metabolism could signal perturbations in multiple biological pathways, including inflammation, mitochondrial function, cell signaling, cell death, endolysosomal trafficking, and exosome biogenesis [27,28,29]. The two lipid species whose abundances have the strongest positive association with AD in the multivariate model are ceramide (24:1) and cholesterol ester (20:1), which are consistent with studies finding an increased abundance of long chain ceramides and cholesterol in AD [30,31,32,33]. A decrease in concentration of four species of phosphatidylethanolamine (PE) were found to be associated with AD in our multivariate model (Table 2). These bioactive glycerophospholipids have been previously associated with AD and PD [34,35], though there is conflicting evidence of decreased PE levels in AD [36], or unchanged between AD and age-matched controls [37].

Several of the targeted metabolites driving our predictive multivariate models have been previously reported in association with AD or PD. Alanine [16,38,39,40], kynurenine [14], glycine [41], tryptophan [40,42], xanthine [43], and serine [40,43] have all been found to be associated with PD, and appear in either our multivariate analysis of the targeted profile, or the pathways and set enrichment analysis derived from univariate analysis. Ornithine, an amino acid involved in the urea cycle, is the metabolite most positively associated with PD. It has been reported to be increased in urine and serum from PD patients [16,44,45] but decreased in CSF of PD patients compared to controls [46]. In our sample, ornithine is strongly associated with PD in the untargeted profile as well, as mummichog identified the urea cycle/amino acid group metabolism as the most enriched pathway for classification between PD patients and controls. Alanine is one of the metabolites most positively associated with AD in our multivariate model (Table 1). Increased levels of this amino acid, as well as several others including glycine, methionine, threonine, phenylalanine, and citrulline, have been found to be increased in CSF of AD patients [47,48,49]. Targeted aqueous metabolomic analysis of premortem blood and postmortem brain from individuals with AD and healthy controls found acylcarnitine to be associated with AD and cognitive changes [50]. We found decreased levels of creatine to be associated with AD in our multivariate model, and proteomic studies have found creatine kinase to be associated with AD patients [51,52]. The positive association between fructose and AD in our multivariate model might provide support for the hypothesis in Johnson et al. [53], suggesting that AD can be driven by an overactivation of fructose metabolism.

Our pathway and set enrichment analysis identified several putative pathways previously associated with AD and/or PD, including biopterin metabolism, which is important for dopamine synthesis, and glutathione metabolism, which is critical for antioxidant activity and mitochondrial integrity. Urea cycle and amino group metabolism were also identified, which are important for fatty acid metabolism, and have been implicated in neuronal function and neurodegeneration. Finally, alanine and asparagine metabolism may be important for metabolic control of neurons [54]. The emergence of these pathways from our analysis suggests that our model correctly predicted PD compared to controls based on changes in metabolites and lipids that arise from significant alterations in cell function, and further supports this method in identifying possible biomarkers as well as biological mechanisms underlying disease pathogenesis.

### Modeling Limitations

There are several limitations to the current study, which we address here. We focus in particular on the near perfect ability to discriminate PD from both controls and AD, which might be met with some skepticism. First, it is possible that our results could be distorted by the effects of dopaminergic medication on the metabolome. While the LEDD was recorded for most PD subjects, it was not used in this study because non-zero LEDD itself is a nearly perfect predictor of PD status. Levodopa treatment has been shown to impact kynurenine, tryptophan, and tyrosine metabolisms [4], which could explain several of the associations found in MSEA (Table 5) and multivariate analysis (Table 2). That said, there is evidence to suggest that the classification is not driven solely by the presence of drugs; we find that a Gaussian elastic net model is unable to predict LEDD better than chance in the PD cohort, and we also find that removing all untargeted metabolites within 1 *m*/*z* of levodopa, entacapone, and their drug metabolites does not change the near-perfect discrimination of PD against controls.

Another potential confounder driving the near perfect PD classification results could come from differences in sample processing. The PD samples were collected by a different study and researchers (Pacific Udall Center) than the AD and control samples (VA MIRECC; see Materials and Methods), and our models were able to classify PD against both AD and controls with high degrees of accuracy. It is possible that, despite strict adherence to the same protocol, there were still differences in sample collection and processing between the two studies that our models were able to detect. One important limitation of our PD analysis is that CSF was extracted at an earlier date for the AD and control subjects than for the PD subjects (Figure A3), and there is some evidence to suggest that prolonged storage time can lead to significant alterations in metabolomic profiles [55]. Determining the relationship between metabolomic profiles and storage time is an active area of research, and we refer to Section 2.5 of [56] for a review of the current literature. However, the exploratory analysis presented in Figure A8 provides evidence to suggest that batch effects are not the sole source of predictive power in our models. In short, the metabolites distinguishing AD and PD are distinct from those distinguishing AD and controls, suggesting that the sample extraction date is not the sole confounder driving these predictions. In addition, studies such as Luan et al. [42] and Willkommen et al. [15] achieve similar performance in distinguishing PD from controls, which indicates that it is possible that the nearly perfect classification in our models is due to a true signal in the metabolome. Furthermore, we note that this analysis is meant to explore the predictive power of the metabolome to distinguish phenotype without relying on known associations with age and sex. As such, the choice to detrend and exclude sex and age in the elastic net models was a conservative choice that shrinks the possibility that the predictive performance and metabolites retained were dependent on age and sex. An approach that explicitly includes age and sex would likely generate more accurate predictions.

Relatedly, the pattern of missing values in metabolomic and lipidomic profiles differed between PD and controls but not between AD and controls. A re-analysis of PD and control metabolite profiles suggests that differential missingness is not solely responsible for the predictive performance of our models; after removing nearly all metabolites that were missing in either or both groups, we still found significant differentiation between PD and controls.

More generally, a larger sample size and further investigation using experimental model systems are required to test whether the discriminatory features that we identified between AD, PD, and controls may have causal biological relationships. In particular, our exploratory analysis of the models distinguishing PD from controls shows that there is a multiplicity of good models, suggesting that the results shown in this study are neglecting potentially important predictors. While the elastic net classification models used in this study are powerful tools for interpretable high dimensional analysis, performing statistical inference (e.g., obtaining *p*-values for individual coefficients) in this setting is an active area of research and is not yet well understood (see [57] for a recent and thorough treatment of this topic). As such, the results of this analysis should be seen as exploratory and as potential directions for future work, rather than making strong inferential claims on its own. Additionally, the null result of being unable to classify GBA variant carriers against non-carriers within the PD cohort is likely due to the small sample size. Specific lipidome alterations in GBA variant carriers with PD have been identified in CSF [58], but these observations were not replicated in our study.

A third limitation of the study is the cross-sectional nature of our data. The metabolome is known to be dynamic and volatile, so patterns identified in a cross-sectional sample might not generalize well. Longitudinal studies, similar to those done by Huo et al. [50] and LeWitt et al. [43], will be needed to determine whether the discriminatory features that we identified between PD, AD, and control CSF are not only diagnostic, but change over time to reflect disease progression.

In conclusion, we identified a number of metabolites and developed an elastic net regression model that can distinguish between AD, PD, and healthy control CSF metabolite profiles, as well as PD versus control lipidomic profiles. Our study is one of the few studies to examine targeted and untargeted aqueous metabolite profiles as well as lipidomic profiles in the two most common neurodegenerative diseases, AD and PD, as well as healthy controls. While our study has some limitations, including small cohort size and lack of longitudinal data, our analyses suggest several directions for future investigation of lipidomic and metabolomic profiles as markers of disease state and progression.

## 4. Materials and Methods

Our analysis was based on cerebrospinal fluid (CSF) samples of 85 healthy subjects, 57 AD subjects, and 56 PD subjects, previously utilized and described in [20]. Sex at birth and APOE variant is known for all subjects, while Glucosylceramidase Beta variant (GBA) and Levodopa Equivalent Daily Dosage (LEDD) (calculated following [59]) are also known for the PD cohort. All samples were collected under University of Washington or VA Puget Sound Health Care System IRB-approved protocols. Written informed consent was given by all participants, or their legally authorized representative, before any study procedures took place. The VA Northwest Mental Illness Research, Education, and Clinical Center (MIRECC) Sample and Data Repository provided the samples for AD and control subjects, while the Pacific Udall Center provided the PD samples used for this analysis.

Samples were collected while subjects were in the fasting state between 0900–1100 h, after the subjects were in supine bedrest for at least 40 min. CSF was extracted via negative pressure using a Sprotte 24 g atraumatic spinal needle, while subjects were placed in the lateral decubitus position. At the bedside, CSF was collected into sterile polypropylene syringes, placed into 0.5 mL aliquots in polypropylene cryotubes, and frozen immediately on dry ice. All samples were stored at −80 ${}^{\circ }$C prior to assay. Figure A3 displays the distribution of the date of LP draws. All participants with AD met the NINDS-ADRDA criteria for probable AD. A summary of the neuropsychological testing results is available in Table 1.

All PD patients were enrolled in the Pacific Udall Center Clinical Core [60]. All PD subjects underwent physical, neurological, and neuropsychological assessments. which are summarized in Table 1, and described in detail in [61]. Cognitive and motor diagnoses were made at consensus conferences, which included at least two movement disorders specialists, a neuropsychologist, and study support personnel. For more detailed information about sample collection for the PD cohort, see [62].

### 4.1. Metabolomics

Details of metabolite processing can be found in previous analyses [20,63]. In brief, we used a LC-MS/MS platform to carry out both global and targeted metabolomics analysis. An Agilent 1200 LC system and a 6520 Q-TOF mass spectrometer (Agilent Technologies, Santa Clara, CA, USA) were used to perform global metabolomic analysis. We prepared 200 μL CSF samples by dissolving in 1000 μL methanol before vortexing and incubating at 20 ${}^{\circ }$C. After centrifuging at 18,000 *g* for 20 min, we collected 750 μL of the supernatant, which was dried and reconstituted in a 100 μL solution of 40% $H_{2}O$: 60% acetonitrile. We then analyzed the samples in positive and negative ESI modes using 5 μL and 10 μL injection volumes, respectively. We performed the chromatographic separation using a Waters XBridge BEH Amide column (15 cm × 2.1 mm, 2.5 μm) (Waters Corporation, Milford, MA, USA) heated to 40 ${}^{\circ }$C, using a flow rate of 0.3 mL/min, 10 mM ${\mathrm{NH}}_{4}{\mathrm{HCO}}_{3}$ in 100% $H_{2}O$ (mobile phase A) and 100% acetonitrile (mobile phase B) with a gradient of 95–10% B from 0 to 5 min, 10% B from 5–40 min, 10–100% B from 40–45 min, and 100% from 40–70 min. We calibrated the Q-TOF/MS before each batch run, and achieved mass accuracy of <1 ppm using a G1969-85000 Agilent Technologies tuning mixture. We then used Agilent Mass Hunter and Mass Profiler Professional to process the MS data. We used an *m*/*z* scan range of 100–2000 and acquisition rate of 1.0 spectra/s.

For targeted analysis, a Sciex 6500+ LC-MS/MS system (Framingham, MA, USA) was used to target 203 known metabolites across more than 25 metabolomic pathways, along with 33 stable-isotope interal standards from Cambridge Isotope Laboratory, Tewksbury, MA, USA. We injected 2 μL of each sample for positive ESI ioniziation mode analysis and 10 μL for negative mode. In both cases, we conducted chromatographic separations using hydrophilic interaction chromatography, employing a Waters XBridge BEH Amide column as before, with flow rate of 0.3 mL per minute, a column compartment temperature of 40 ${}^{\circ }$C, and autosampler temperature of 4 ${}^{\circ }$C. We used 5 mM ammonium acetate in $H_{2}O$ with 0.5% acetic acid and 0.5% acetonitrile (Solvent A) and acetonitrile with 0.5% acetic acid and 0.5% water (Solvent B) to compose the mobile phase. For both negative and positive ionization modes, we used the following gradient: isocratic elution of 10% A for 0–1.5 min, a linear increase of A to 65% from 1.5–9 min, constant 65% A from 9–14 min, and 10% at 15 min. We used authentic commercially obtained compounds (Fisher Scientific, Pittsburg, PA, USA or Sigma-Aldrich, Saint Louis, MO, USA) to develop the assays, and MultiQuant 3.0.2 was used to obtain metabolite concentrations, as described previously [64].

Details of lipidomic analysis have been described previously [20,65,66]. For lipidomic analysis, we added 54 isotope labeled internal standards (spanning 13 lipid classes) purchased from Sciex. We thawed the frozen CSF samples at 25 ${}^{\circ }$C for 30 min, then vortexed and transferred 100 μL of CSF to a borosilicate glass culture tube. We extracted lipids using dichloromethane/methanol, then concentrated the extracts under nitrogen, and reconstituted the samples in 100 μL of 10 mM ammonium acetate in dichloromethane:methanol (50:50). We analyzed the resulting lipids using the Sciex Lipidyzer platform: specifically, we used a Shimadzu LC and AB Sciex QTRAP 5500 MS/MS system, where differential mobility spectrometry (DMS) is conducted using SelexION. We then used multiple reaction monitoring to quantify lipids in both positive and negative ionization modes, with and without DMS, before processing the data using Sciex Analyst 1.6.3 and Lipidomics Workflow Manager 1.0.5.0. The Lipidizer platform outputs an abbreviated lipid annotation that does not exactly match the LIPID MAPS shorthand annotation [67]. Table S1A of [66] provides a table of translation between the two shorthand notations.

As described in [64], we monitored instrument performance over long periods and assisted with normalization by running both a pooled study Quality Control (QC) sample and a commercially pooled lab QC sample once every 10 study samples, and at the beginning and end of all sample batches. The pooled study QC sample is made by combining 10–20 μL of each sample, which are prepared as usual alongside the study samples.

### 4.2. Preprocessing

Samples were prepared for assay in seven batches of 28 or 29 samples each. As described in [20], batch effects are reduced using the finite selection model from [68] to balance age, sex, APOE status, and disease status across the batches. An XGboost model (a nonparametric regression method implemented in the ‘xgboost‘ R package) was fit separately on each metabolite using run index, age, and sex as covariates. Cross validation was used to select tuning parameters for these models [69]. The residuals of this procedure were used for the remainder of the analysis, which reduces the likelihood that the predictive performance of our models solely reflects metabolite drift over time, or the known associations of age and sex with both AD and PD [70,71,72]. The inclusion of run index in this process supplements the QC sample normalization done in the metabolite processing, and can be helpful, especially when considering metabolite drift between QC runs. The benefit of using a flexible tree-based method like XGboost is that it is capable of handling and correcting nonlinear and discontinuous metabolite drift. Examples of this drift correction procedure in our dataset are shown in Supplementary Figure 2 of [20].After this correction, each metabolite was centered and scaled to have unit variance. Within each feature, abundances further than three Median Absolute Deviations (MAD) were removed and treated as missing values to limit the influence of unreasonably small or large abundances, which are most likely the result of processing noise. To limit the impact of missing data imputation on the analysis, we excluded features with more than 50% missingness from this analysis. The remaining missing abundances were imputed using multiple imputation, whereby multiple (in our case, 5) versions of imputed data are created. Identical analysis is done on each of the five imputed datasets, which provides a measure of the variability induced by the missing data procedure. The Amelia package in R [73] was used to perform this procedure, with details of the implementation to accommodate large metabolomic datasets available in [20]. Missing data imputation was done within each cross validated loop (described in Section 4.3) to avoid the imputation depending on information from both the training and testing sets.

### 4.3. Regression Modeling

To model the extent to which our data discriminate AD and PD against controls, we fit regularized logistic regression models, which extend the classic logistic regression model to allow for the number of features in our dataset to exceed the number of rows. In particular, we used elastic net regression, which fits
(1)$$\min_{\beta_{0},\beta_{1}\in R^{1+p}}-\left[\frac{1}{n}\sum_{i=1}^{n}y_{i}\left(\beta_{0}+x_{i}^{⊤}\beta \right)-\log\left(1+e^{\beta_{0}+x_{i}^{⊤}\beta }\right)+\lambda \left(\frac{{\left(1-\alpha \right)∥\beta ∥}_{2}^{2}}{2}+\alpha {∥\beta ∥}_{1}\right)\right]$$

Here, α is a value between 0 and 1 which determines the type of regularization, while λ is a non-negative value that determines the total amount of regularization. We used the R package glmnet to fit these models, choosing α=0.5 and selecting the value of λ that resulted in the smallest deviance in leave one out cross validation [17]. This choice of α allowed for variable selection without imposing a strict upper limit on the number of selected variables. To mitigate the effects of class imbalance, observations were weighted by inverse proportion of their class label. For interpretability, in Table 2, Table 3 and Table 4, we show exponentiated coefficients for models fit on the full data (for the targeted and lipidomic profiles) to classify AD against controls, PD against controls, and AD against PD, respectively. Because the covariates are standardized, these exponentiated coefficients correspond to the expected odds ratio (OR) resulting from a standard deviation increase in metabolite/lipid concentration.

To estimate out of sample predictive power, leave one out prediction was performed, i.e., missing data imputation and regression modeling were iteratively done on the training data after removing a single observation, and a prediction was formed on the left out observation. Rather than imputing missing values on a left out observation, missing features were removed from that iteration’s test set, using a process known as reduced feature modeling [74]. These left out predictions were used to create Receiver Operating Characteristic (ROC) curves and compute Area Under the ROC Curve (AUC). To summarize the performance of our models, we compute the average AUC value across all five missing data imputations for each leave-one-out iteration.

### 4.4. Pathway Analysis

To perform pathway analysis and metabolite set enrichment analysis, we construct univariate classification models by separately regressing each metabolite against disease status, which allows for correlated and collinear metabolites to be included in their respective pathways. For univariate untargeted metabolomic analysis, we used logistic regression models to perform the classification, and then inputted mass:charge ratios, retention times, *t*-values, and Benjamini–Hochberg corrected *p*-values into Mummichog via a Python script [75] to perform pathway analysis. Positive and negative ionization modes were analyzed separately.

For univariate targeted analysis, we used the same logistic regression model technique, and inputted the names of all metabolites with false discovery rate <0.05 into Metabolite Set Enrichment Analysis (MSEA), which performed hypergeometric tests of associations with known metabolite sets [76]. To perform MSEA on the targeted dataset, we used the open source MetaboAnalystR package [77].

## Figures and Tables

**Figure 1 metabolites-12-00277-f001:**
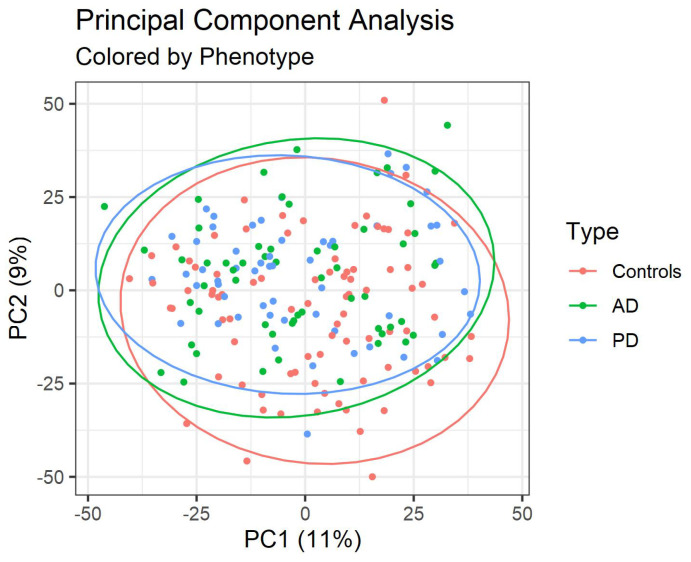
Untargeted data projected onto the first two Principal Components (PC). Each point represents a subject, colored by their phenotype. Percentages in the axis titles refer to the percentage of variation of the data explained by the respective PC. In addition, 95% confidence ellipses assuming the *t*-distribution are also plotted. The first two principal components do not clearly separate the disease phenotypes.

**Figure 2 metabolites-12-00277-f002:**
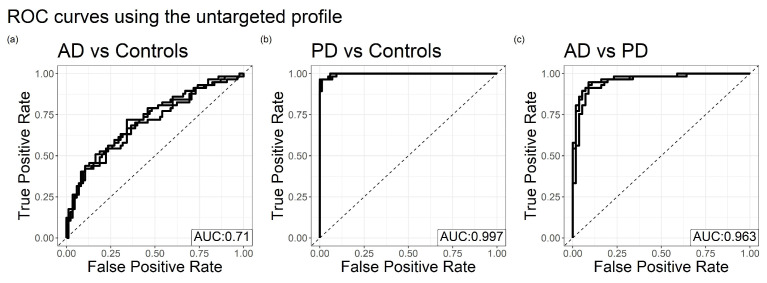
Receiver Operating Characteristic (ROC) Curves for binomial elastic net regressions classifying (**a**) controls against subjects with AD, (**b**) controls against subjects with PD, and (**c**) subjects with AD against subjects with PD. Solid lines represent models formed using each of the five missing data imputed datasets. The dotted y=x line represents the ROC curve under a model which makes predictions at random. The average Area Under the Curve (AUC) across the five ROC curves is displayed in the bottom right.

**Figure 3 metabolites-12-00277-f003:**
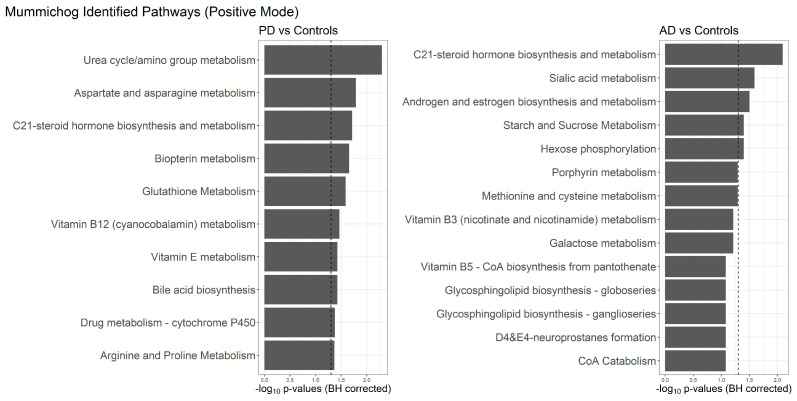
Pathway Analysis of Mummichog on the positive mode untargeted metabolites from univariate logistic models classifying PD and AD against Controls, sorted by $-\log_{10}$ Benjamini–Hochberg corrected *p*-values, with the vertical dashed line marking p=0.05.

**Table 1 metabolites-12-00277-t001:** Summary of subject data split by phenotype (top), with cognitive test results (middle), and with additional PD subject information (bottom). The cognitive status for PD was classified at three levels: no cognitive impairment, mild cognitive impairment (MCI), and dementia. Age of onset refers to the age of onset of motor symptoms. GBA refers to the carrier frequency for pathogenic GBA mutations and the E326K polymorphism. A chi-squared test of independence for sex and phenotype reports a *p*-value of 0.051. A one-way ANOVA of age at time of LP and phenotype reports a *p*-value of $2.099\times 10^{-9}$ [18].

	Control	AD	PD	
n	85	57	56	
Age at time of LP	53.7±20.3	70.6±9.8	65±10	
Duration of disease	N/A	4.3±2.8	9±5	
ApoE genotype	2.3 (9%) 2.4 (1%) 3.3 (53%) 3.4 (33%) 4.4 (4%)	2.3 (3.5%) 2.4 (3.5%) 3.3 (44%) 3.4 (33%) 4.4 (16%)	2.2 (1.8%) 2.3 (14.3%) 3.3 (50%) 3.4 (26.8%) 4.4 (7.1%)	
Race (% white)	91.7%	94.7%	94.6%	
Sex	53% M (41 F)	49% M (29 F)	70% M (17 F)	
	**Control**	**AD**	**PD**	
MMSE total score (0–30)	29.4±0.9	21.4±5.6	N/A	
Logical memory immediate recall (0–25)	13.5±4.0	5.2±13.0	12.6±3.0	
Category fluency (animals) (0–999)	23.3±6.3	9.7±5.1	20.6±5.3	
Trail Making Test Part A (s) *	25.4±2.0	107.3±182.1	30.9±12.2	
Trail Making Test Part B (s) *	68.1±34.1	406.4±357.3	76.9±43.5	
Logical memory delayed recall (0–25)	12.5±4.1	3.4±13.1	11.7±3.6	
	**All**	**No Cognitive Impairment**	**MCI**	**Dementia**
n	56	16	36	4
Sex	70% M (17 F)	56% M (7 F)	70% M (11 F)	100% M
Race (% white)	94.6%	100%	91.7%	100%
Age of onset of motor symptoms	56±10	52±11.1	57±9	58±16.6
Age at time of LP	65±10	60±9.1	67±8.6	65±21.4
Duration of disease	9±5	8±5.2	10±4.9	7±5.3
Levodopa equivalent dose	714±536	441±324.9	802±592.3	1029±221.7
MDS-UPDRS III	25±12.8	21±11.4	24±12.4	38±17.6
Hoehn & Yahr stage	2±0.6	2±0.4	2±0.5	3±1.3
MoCA	25±3.8	27±2.4	25±2.1	16±5.4

* For PD subjects, Trail Making Test Part A was truncated at 150 s, and Part B was truncated at 300 s.

**Table 2 metabolites-12-00277-t002:** Names and Odds Ratios (OR) for targeted metabolites and lipids retained in all five imputations of elastic net models fit on the full data to classify AD patients against controls. The tables are sorted by magnitude and split into positive and negative coefficient tables. Because the features were standardized prior to fitting the models, these ORs represent the expected odds ratio resulting from a standard deviation increase in concentration. Only ORs with magnitude greater than 1.1 or less than 0.9 are shown here. A two standard deviation interval is shown for the OR to quantify variability across the five missing data imputations.

Targeted Metabolites–Positive Coefficients	
**Metabolite**	**OR ± 2SD (>1)**
1-Methyladenosine	1.52 (1.43, 1.62)
Glycine	1.38 (1.3, 1.46)
Alanine	1.38 (1.32, 1.43)
Sarcosine	1.21 (1.16, 1.26)
Acetylcarnitine	1.19 (1.16, 1.21)
4-Methoxyphenylacetic acid	1.17 (1.11, 1.25)
Sorbitol	1.15 (1.13, 1.17)
Lactate	1.14 (1.12, 1.16)
Hydrocortisone	1.14 (1.09, 1.19)
Homoserine	1.12 (1.07, 1.16)
Caffeine	1.11 (1.04, 1.17)
**Metabolite**	**OR ± 2SD (<1)**
*N*-Acetylneuraminic acid	0.76 (0.71, 0.80)
Glycocyamine	0.80 (0.79, 0.82)
4-Aminobutyric acid	0.84 (0.81, 0.88)
Creatine	0.85 (0.80, 0.90)
Urocanic acid	0.86 (0.73, 1.01)
Homocysteine	0.88 (0.84, 0.91)
Uridine	0.89 (0.85, 0.92)
**Lipids—Positive Coefficients**	
**Lipid**	**OR ± 2SD (>1)**
SM(18:1)	1.51 (1.42, 1.60)
CE(16:1)	1.22 (1.20, 1.25)
CE(20:1)	1.19 (0.91, 1.54)
PC(18:0/20:3)	1.12 (0.99, 1.26)
**Lipids—Negative Coefficients**	
**Lipid**	**OR ± 2SD (<1)**
PE(P-18:0/22:6)	0.77 (0.76, 0.79)
PE(18:0/20:4)	0.84 (0.79, 0.89)
PE(18:0/22:6)	0.90 (0.84, 0.98)

**Table 3 metabolites-12-00277-t003:** Names and OR for targeted metabolites and lipids retained in all five elastic net models fit on the full data to classify PD patients against controls. Only ORs with magnitude greater than 1.1 or less than 0.9 are shown here. A two standard deviation interval is shown for the OR to quantify variability across the five missing data imputations.

Targeted Metabolites—Positive Coefficients	
**Metabolite**	**OR ± 2SD (>1)**
Ornithine	2.10 (1.82, 2.41)
Glycylproline	1.75 (1.52, 2.01)
Levulinic acid	1.62 (1.43, 1.82)
Acetylglycine	1.57 (1.42, 1.73)
Glycine	1.57 (1.45, 1.70)
Creatinine	1.52 (1.46, 1.58)
Cytosine	1.48 (1.28, 1.70)
Adenosine	1.45 (1.26, 1.67)
Pentadecanoic acid	1.40 (1.32, 1.49)
Sorbitol	1.40 (1.30, 1.52)
*N*-Acetylethanolamine	1.39 (1.31, 1.48)
alpha-Hydroxyisovaleric acid	1.39 (1.23, 1.57)
2-aminoadipic acid	1.36 (1.16, 1.60)
Methylguanidine	1.32 (1.27, 1.38)
Xanthosine	1.25 (1.20, 1.30)
Dimethylarginine	1.22 (1.15, 1.30)
Homoserine	1.21 (1.14, 1.28)
Threonine	1.20 (1.15, 1.25)
Cystine	1.16 (1.09, 1.23)
3α-Hydroxy-12 Ketolithocholic Acid	1.16 (1.09, 1.23)
Adenosyl-l-homocysteine	1.15 (1.08, 1.22)
6-Methyl-dl-tryptophan	1.13 (1.06, 1.20)
Anthranilic acid	1.12 (1.03, 1.21)
Fructose	1.11 (1.02, 1.20)
**Targeted Metabolites—Negative Coefficients**	
**Metabolite**	**OR ± 2SD (<1)**
Indole-3-acetic acid	0.57 (0.54, 0.61)
Serine	0.58 (0.52, 0.64)
*N*-Acetylneuraminic acid	0.61 (0.55, 0.68)
Urocanic acid	0.64 (0.53, 0.76)
Agmatine	0.65 (0.63, 0.68)
HIAA	0.66 (0.60, 0.73)
Glycocyamine	0.71 (0.58, 0.87)
Aspartic acid	0.76 (0.66, 0.88)
4-Methylvaleric acid	0.79 (0.73, 0.85)
Serotonin	0.82 (0.77, 0.87)
Mannose	0.82 (0.74, 0.90)
Creatine	0.83 (0.78, 0.88)
Xanthine	0.83 (0.76, 0.90)
4-Aminobutyric acid	0.86 (0.81, 0.91)
4-Methoxyphenylacetic acid	0.86 (0.81, 0.91)
Citraconic acid	0.87 (0.74, 1.02)
Decanoylcarnitine	0.89 (0.84, 0.94)
**Lipid**	**OR ± 2SD (>1)**
PE(P-16:0/18:1)	1.54 (1.45, 1.63)
HCER(18:0)	1.49 (1.43, 1.55)
FFA(16:1)	1.46 (1.30, 1.65)
SM(18:1)	1.42 (1.26, 1.60)
FFA(24:0)	1.22 (1.11, 1.35)
PC(16:0/20:2)	1.21 (0.93, 1.57)
FFA(20:2)	1.20 (0.94, 1.52)
CE(20:1)	1.20 (0.98, 1.46)
DAG(20:0/20:0)	1.17 (0.96, 1.43)
PE(16:0/22:6)	1.16 (0.97, 1.39)
LPC(18:1)	1.11 (1.06, 1.15)
**Lipids—Negative Coefficients**	
**Lipid**	**OR ± 2SD (<1)**
PC(18:1/18:2)	0.49 (0.45, 0.53)
FFA(18:0)	0.64 (0.53, 0.76)
PE(18:1/18:1)	0.65 (0.48, 0.88)
FFA(24:1)	0.68 (0.61, 0.75)
PC(18:1/20:4)	0.72 (0.64, 0.81)
PC(18:0/22:6)	0.76 (0.70, 0.82)
PC(18:1/16:1)	0.88 (0.79, 0.97)

**Table 4 metabolites-12-00277-t004:** Names and OR (associated with a standard deviation increase in concentration) for targeted metabolites retained in all five elastic net models fit on the full data to classify PD patients against AD patients. A metabolite with OR >1 indicates that higher concentration is associated with AD in our models. Only ORs with magnitude greater than 1.1 or less than 0.9 are shown here. A two standard deviation interval is shown for the OR to quantify variability across the five missing data imputations.

Targeted Metabolites—Positive Coefficients	
**Metabolite**	**OR ± 2SD (>1)**
Serine	1.63 (1.36, 1.95)
Alanine	1.62 (1.46, 1.79)
Indole-3-acetic acid	1.52 (1.46, 1.58)
Xanthine	1.42 (1.26, 1.60)
Aspartic acid	1.40 (1.32, 1.49)
Caffeine	1.40 (1.30, 1.52)
Sarcosine	1.22 (1.15, 1.30)
HIAA	1.20 (1.11, 1.30)
*N*-glycyl-l-proline	1.16 (1.01, 1.34)
Glycodeoxycholic acid	1.15 (1.06, 1.25)
4-Methoxyphenylacetic acid	1.14 (1.12, 1.16)
Serotonin	1.13 (1.08, 1.17)
**Targeted Metabolites—Negative Coefficients**	
**Metabolite**	**OR ± 2SD (<1)**
Ornithine	0.52 (0.51, 0.53)
alpha-Hydroxyisovaleric acid	0.63 (0.58, 0.68)
Homocysteine	0.64 (0.59, 0.70)
Histidine	0.70 (0.62, 0.79)
Creatinine	0.72 (0.66, 0.78)
Glycylproline	0.73 (0.64, 0.82)
Levulinic acid	0.76 (0.70, 0.83)
Adenosine	0.77 (0.67, 0.89)
*N*-Acetylethanolamine	0.81 (0.75, 0.88)
Acetyl-l-glutamine	0.86 (0.79, 0.93)

**Table 5 metabolites-12-00277-t005:** Names and OR for targeted metabolite abundance missingness indicators in AD classification against controls, lipid abundance missingness indicators in PD classification against controls, and lipid abundance missingness indicators in AD classification against PD, retained in elastic net models fit on the full data. The OR corresponds to increased odds of classifying a patient as having AD/PD/AD associated with a standard deviation increase in metabolite concentration (for left, middle, and right, respectively). Age and sex are not penalized, and are therefore guaranteed to be included in these models. Other combinations of phenotype and profile (i.e., targeted-PD or lipids-AD tables) are not shown because the only retained covariates were age and sex.

Targeted Metabolites—AD v C	
**Metabolite**	**OR (AD v C)**
Citraconic acid	0.53
Phenylalanine	1.85
Creatinine	1.59
Glucosamine	1.46
Amiloride	1.42
*N*-Acetylneuraminic acid	0.71
Mannose	0.73
Male	0.85
Age	1.07
Creatine	1.02
**Lipids—PD v C**	
**Lipid**	**OR (PD v C)**
Male	1.85
TAG46:0-FA16:0	1.77
DAG(18:1/22:6)	1.08
Age	1.04
**Lipids—AD v PD**	
**Lipid**	**OR (AD v PD)**
Male	0.25
PE(18:1/18:1)	0.68
PC(16:0/14:0)	0.74
TAG52:4-FA16:1	1.17
CE(18:4)	1.10
Age	1.09

## Data Availability

Data used in this analysis are publicly available in FigShare at doi.org/10.6084/m9.figshare.14816622.v3.

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
