# Peer review of "Predictive Modeling of Alzheimer’s and Parkinson’s Disease Using Metabolomic and Lipidomic Profiles from Cerebrospinal Fluid"

_metabolites, 2022, doi:10.3390/metabo12040277_

Round 1
Reviewer 1 Report
In their manuscript entitled “Predictive modeling of Alzheimer’s and Parkinson’s disease using metabolomic and lipidomic profiles from cerebrospinal fluid” (manuscript-ID metabolites-1586838), Nathan Hwangbo and co-authors describe a study on metabolomics in Alzheimer’s disease (AD), Parkinson’s disease (PD), and healthy age-matched controls. The authors analyzed targeted and untargeted aqueous and lipidomic profiles of the metabolome from human cerebrospinal fluid CSF) to build multivariate predictive models distinguishing the three different groups as indicated. The authors found separation in the metabolome between PD and controls, as well as between PD and AD, with weaker separation between AD and controls with alanine, kynurenine, tryptophan, and serine being associated with PD compared to controls, and alanine, creatine, and long chain ceramides being associated with AD compared to controls. In a univariate pathway analysis of untargeted and targeted metabolite profiles they found vitamin E and urea cycle metabolism pathways being associated with PD, and the aspartate/asparagine as well as c21-steroid hormone biosynthesis pathways being associated with AD. Finally, they found the amount of metabolite missingness to vary by phenotype and discussed the importance of examining missing data in future metabolomic studies.
To my opinion, this is an excellent study combining robust participant examination and study conduction as well as sophisticated laboratory and statistical methods. Finally, the manuscript is written well in all its parts. I do not have any concerns.
Author Response
Thank you for the kind words!
Reviewer 2 Report
The manuscript submitted by Hwangbo et al. summarizes a study focused on the metabolomics (targeted and untargeted) and lipidomic analysis of the cerebrospinal fluid in Alzheimer’s and parkingson’s disease. The manuscript is well written and it is easy to understand. The authors have clearly mentioned the background and rationale of the study together with limitations. In my opinion, it is an interesting approach for the identification of relevant biomarkers taking into account sex and age of patients and deserve publication. However, some points provide clarifications.
- Is there a difference in storage time between controls+AD subject and PD patients before analyses? If any, it could be mentioned and considered in discussion.
- In table 1, there are clearly several differences between groups on age, sex, etc. It will be interesting to add some statistical analysis of the different variables; it will justify the batch normalisation performed by authors further. Please, transpose or simplify it to make it clearer.
- Metabolomics section. For the untargeted approach, the flow rate is lacking. Please add briefly some information on the identification process for the targeted analysis (MS/MS experiments, delta ppm cut-off, standards used, etc.) as well as for the lipidomic part. In this last one, more information about the extraction will be appreciate (ie. Amount of sample)
- Preprocessing section. I am not familiar with the finite selection model used by authors, but I appreciate that confounding effects as run index, age and sex seems to be correctly considered. However, in metabolomics and lipidomics QC (quality control, generally a mix of all samples) are generally use for this purpose; authors did not mention them. Please clarify this part.
- Lipid annotation. Please, update or check your lipid annotation according to your level of identification (doi: 1194/jlr.S120001025)
- Table 3: please verify the spelling of the hydroxyl-ketolithocholic acid
- Pathway and set enrichment analysis. Several pathways are highlighted, how many metabolites of the targeted approach are found from these pathways? Are results coherent? It could be very interesting to further discuss on them.
Reviewer 3 Report
In this paper, Hwangbo et al. aimed at developing a prediction model to discriminate Alzheimer and Parkinson patients from healthy controls, exploiting metabolomic and lipidomic data. Unfortunately, I do not consider the paper suitable for publication, as it suffers from many methodological issues which impact on results. Below, a list of major and minor concerns.
Major:
1) The ‘Preprocessing’ section is really confusing and some steps are completely unclear: XGboost is generally used to perform a features selection; what is the purpose here? Moreover, the authors removed data points with high variability and then re-assigned them a value (imputation), introducing noise. Did you try what happens without imputation?
2) In the ‘Regression modelling’ section, the authors (correctly) explain that their dataset is imbalanced. However, they assessed the prediction performances with ROC curves; Precision-Recall curves are better in these cases. Moreover, sensitivity, specificity, PPV, NPV and F1 score are requested.
3) In Tables 2,3,4,5 the authors showed the OR values for relevant predictors. The corresponding p-values should be reported. Moreover, it can be useful to assess how the OR changes over leave-one-out iterations (i.e. averaged OR +- sd).
4) In the penalized regressions, Age and Sex should be used as blocking factors, rather than adjusting data. Moreover, are there other potential covariates?
5) The pathways analysis cannot be performed after a classification/prediction analysis but after a standard statistical differential analysis, in order to get features (even correlated/collinear) belonging to the same pathways. The higher the number of DE features belonging to the same pathway, the higher the significance of the pathway.
Minor:
1) Table 1 must be transposed, reformatted and p-values (from ANOVA or Chi squared) must be provided, when appropriate.
2) Figure 1 is not a result, rather diagnostic plots; so, should be moved in supplementary.
3) The authors exploited an ‘Elastic Net’ to select relevant features and perform the classification step; however, the title of section 3.1. introduces the PLS-DA. The authors did not implement this kind of regression analysis.
4) In Figure 3, there are many ROC curves in each panel; I think they represent the ROC for each LOO iteration or dataset. However, there is just one AUC value; is it the averaged value?
Reviewer 4 Report
I am not an expert in metabolomics, so I will leave the pros and cons of the content there to the editors. As for the statistical methods used, I don't see any problems with them. But when developing a model in an exploratory analysis methods, racial bias and small sample size are likely to have a significant impact on generalization. Therefore, the conclusion should be a modest claim. It is also recommended that a flowchart diagram of the analysis be included to facilitate the reader's understanding.
Author Response
Thank you for your comments. We completely agree that the results in are paper should be very modest, given our small and skewed sample size. We have modified the discussion section to emphasize this point further. Per your suggestion, we have also added a flowchart diagram to the appendix to outline the steps of our analysis.
Round 2
Reviewer 2 Report
Authors have taken into account reviewers suggestions and comments. Thanks to these modifications I believe that the manuscript is clearer and deserve publication.
Reviewer 3 Report
I thank the authors for addressing all my concerns and allowing me to contribute to this interesting manuscript..